# Prevalence of HIV and Selected Disease Burden in Outpatients of Primary Health Care (PHC) Facilities in Rural Districts of the Eastern Cape Province, South Africa

**DOI:** 10.3390/ijerph19138003

**Published:** 2022-06-29

**Authors:** Ntombizodumo Nxasana, Kelechi E. Oladimeji, Guillermo-Alfredo Pulido-Estrada, Teke R. Apalata

**Affiliations:** 1Department of Laboratory Medicine and Pathology, Faculty of Health Sciences, Walter Sisulu University, Mthatha 5100, South Africa; zodumonxasana@gmail.com (N.N.); oladimejikelechi@yahoo.com (K.E.O.); 2College of Graduate Studies, University of South Africa, Pretoria 0001, South Africa; 3Department of Public Health, Faculty of Health Sciences, Walter Sisulu University, Mthatha 5100, South Africa; pulido762@yahoo.com

**Keywords:** COVID-19, primary health care, clinical profile, rural, prevailing disease condition, Eastern Cape Province, South Africa

## Abstract

Assessing underlying illnesses can inform health stakeholders about chronic conditions for targeted enhanced prevention and treatment strategies. Since the Eastern Cape Province has a high disease burden, this study aimed to assess the prevalence of human immunodeficiency virus (HIV) infection and selected disease burden of outpatients from primary health care (PHC) facilities in the districts. From February 2019 to February 2021, a cross-sectional study was conducted. Research Electronic Data Capture (REDCap)-enabled tablets were used to collect data from consenting outpatients over the age of 18 years using an interviewer-administered WHO core and expanded stepwise questionnaire. The statistical analysis was mainly descriptive with the use of counts, frequencies, and summary measures. The study population was predominantly female (86.5%). Prevalent diseases included HIV, hepatitis B virus (HBV) infection, and cardiometabolic diseases. HIV prevalence was 52% and highest in the age group of 30–59 years. In people living with HIV, the nonsuppressed viral load (VL 1000 copies/mL) was highest in the age group of 40–49 years (34.6%). Prevalence of diabetes was highest at the Mhlontlo subdistrict (42.3%), while the King Sabata Dalindyebo (KSD) subdistrict had the highest prevalence of HBV infections (39.1%). Based on the findings, we advocate for intermittent assessments of disease burden in certain settings, such as rural areas, to improve PHC practice and outcomes, especially in the wake of the coronavirus disease (COVID-19) pandemic.

## 1. Introduction

Based on available evidence, the burden of diseases is categorized into communicable, noncommunicable, and injuries [1,2]. There is also growing concern about the impact of the epidemiologic and demographic shift from communicable to noncommunicable diseases (NCDs) [3]. Human immunodeficiency virus and acquired immunodeficiency syndrome (HIV/AIDS), tuberculosis (TB), neglected tropical diseases, and malaria, compounded by the ongoing coronavirus disease (COVID-19) pandemic, account for most of the current global burden of communicable diseases [4]. Similarly, cardiovascular diseases account for the majority of NCD burden [5]. In 2019, 55% of the 55.4 million deaths worldwide were caused by the top ten diseases, which include: cardiovascular (ischemic heart disease, stroke), respiratory (chronic obstructive pulmonary disease, lower respiratory infections), and neonatal conditions (birth asphyxia and birth trauma, neonatal sepsis and infections, and preterm birth complications) [6]. Unfortunately, Sub-Saharan Africa bears a quarter of the world’s disease burden despite accounting for only 11% of the world’s population [7]. The most prominent disease burden groups in Sub-Saharan Africa include communicable, noncommunicable, maternal, neonatal, and nutritional diseases [8,9,10].

Presently, South Africa, which is a middle-income country in Sub-Saharan Africa, faces a quadruple burden of disease in the form of communicable diseases, NCDs, maternal and child mortality, as well as injury and trauma [11]. The country has the highest HIV prevalence rate globally and operates the world’s most extensive antiretroviral therapy (ART) program [12]. With 7.8 million people living with HIV in South Africa in 2020, 72% of whom are on ART, the increase in NCDs could lead to a higher disease burden [13]. The primary health care (PHC) is the first level of contact with health systems in the healthcare delivery domain and has recently been recognized as a critical component in achieving sustainable development goals (SDGs) and universal health coverage (UHC) [14,15,16]. According to the world health organization (WHO), scaling up PHC interventions in low- and middle-income countries through high-quality care could save 60 million lives and increase average life expectancy by 3.7 years [17]. However, access to PHC is insufficient to achieve the SDGs and UHC; high-quality care for evidence-based, identified prevailing disease conditions by the health systems is required [16,18].

The Eastern Cape Province (ECP) is one of South Africa’s poorest provinces [19]. Consequently, an unaddressed or overlooked disease burden could strain its population, already stretched socioeconomically, resulting in worsening health conditions. A thorough systematic assessment involving the detection of asymptomatic and coexisting conditions, an accurate diagnosis, appropriate and timely treatment, referral for hospital care and surgery when necessary, patient follow-up, and adjustment of treatment course as necessary are all components of high-quality care [18]. If health systems were aware of specific disease conditions to target with appropriate interventions, they would be able to make the best use of available resources [20]. Thus, it is critical to understand the disease burden profile to provide adequate, high-quality health services, particularly in low-income settings [20,21]. Unfortunately, there is a dearth of published articles and research studies that provide data on prevailing diseases among residents in rural communities within the Eastern Cape. This study aimed to provide evidence by assessing the HIV prevalence and selected disease burden in outpatients from PHC facilities in rural districts of the Eastern Cape Province, South Africa.

## 2. Materials and Methods

### 2.1. Study Design, Period, Setting, and Population

Between February 2019 to February 2021, this cross-sectional descriptive study was carried out in six (6) PHC clinics within selected districts of the Eastern Cape Province. The Eastern Cape is one of the nine (9) provinces in South Africa; it has the fourth highest population of approximately 6, 676, 590, which constitutes 11.1% of the country’s total population [19]. In the ECP, there are 2 metropolitan municipalities and 6 district municipalities which are further subdivided into 31 local municipalities (Figure 1) [22].

### 2.2. Sampling and Population

The selection and enrollment of the study participants was carried out by multistage random sampling. In the first stage, two municipal districts (OR Tambo and Joe Gqabi) were randomly selected from a total of six municipalities by a random draw from a box where all the municipality names were individually written and placed. The second stage, which also used the random draw, involved the random selection of three (3) subdistricts and an additional six (6) PHCs (Figure 2). The last stage involved the random selection of consenting study participants, consisting of outpatients aged 18 years and above that visited the selected PHCs for either symptomatic health concerns, chronic health conditions, or routine clinic visits.

### 2.3. Sample Size Estimation

In the Eastern Cape, the burden of disease in 2018 was such that HIV/AIDS was the second leading underlying natural cause of death with a 5.9% prevalence rate [23]. Therefore, taking into consideration this 5.9% disease burden rate with a 95% confidence level, design effect of 1.5, and margin of error as 2.1%, the minimum sample size for the study was estimated at 722.

### 2.4. Data Collection, Management, Variables of Interest, and Analysis

#### 2.4.1. Data Collection Tool

Data were collected using a validated WHO core and expanded stepwise questionnaire [24], which was modified to suit the context of this study. The main questions in the original version of the questionnaire included questions on sociodemographic data, lifestyle behaviors, and reported general health, such as the medical history of any diseases. In addition to anthropometric measurements, in vitro, bio-physiological measures such as HIV testing, blood sugar levels, and blood pressure were also included. The tool was modified to obtain information on hepatitis B virus (HBV) infection because of a presumed immuno-compromised state of those living with HIV.

#### 2.4.2. How Data Was Collected (Participant Enrolment Procedure)

Data on sociodemographic characteristics, lifestyle behaviors, reported general health, such as the medical history of any diseases, and anthropometrics were primarily collected during the modified WHO core and expanded stepwise questionnaire administration. The questionnaire which was configured on tablets through the research electronic data capture (REDCap) platform was interviewer-administered on consenting, eligible participants. These participants were randomly selected during their clinic visits and approached for informed consent and enrolment into the study. For the in-vitro bio-physiological measures, secondary data on HIV and HBV infection statuses, latest viral loads, and serum lipids were obtained from the patient’s laboratory investigation results documented in the patients’ medical records.

#### 2.4.3. Variables of Interest (Outcomes)

Targeted outcomes for the study included prevailing diseases to be identified in the study population. Sociodemographic characteristics such as age, sex, marital status, income, and health-related variables were used as explanatory variables for identified prevailing disease conditions, such as HIV, and comorbidities (hypertension, diabetes, chronic cardiac disease), HBV status, and viral loads levels for those infected with HIV.

#### 2.4.4. Data Management and Analysis

Data collected on REDCap, an electronic clinical and translational research database, were exported with Microsoft Office Excel to Statistical Package for Social Sciences (SPSS) version 22 for data management. Patient identifiers were anonymized with the use of assigned study IDs for each participant to ensure confidentiality of their information. Variables of interest from the database included sociodemographic and sociobehavioral information (age, gender, marital status, employment, smoking); and clinical information (viral load, coinfections, and comorbidities at baseline). For data analysis, both STATA version 15 and SPSS were used. Continuous and categorical variables were presented using tables and graphs. The 95% Confidence Interval (CI) was used to estimate the precision of estimates. The level of significance was set at 5% (*p*-value < 0.05).

## 3. Results

### 3.1. Sample Description

A total of 796 patients were enrolled in the study and of these, 46 were excluded because of incomplete data. The final study population was 750.

### 3.2. Sociodemographic Characteristics of the Study Population

The study population was predominantly female (86.5%) with half of the respondents single, while 35% were married people. Additionally, 58% of the study population had completed secondary education, while 57% were unemployed (Table 1).

### 3.3. HIV Prevalence by Age and Gender

The mean age for PLHIV was 43.5 years, and 48.4 for HIV uninfected individuals (Figure 3a). Prevalence of HIV infection was higher in the age group of 30–59 years, with the age group of 30–39 demonstrating the highest prevalence at 28.7% (Figure 3b). When the prevalence of HIV was stratified by gender and age groups, a similar trend in HIV infection was observed for both genders across the age categories (Figure 3c). However, males between the ages of 40–49 years were shown to have a lower HIV infection rate of 33.3%, compared to the females who had a 55.3% HIV infection rate (Figure 3c).

### 3.4. The HIV Population Viral Load Status by Age and Subdistrict

Among those with nonsuppressed viral load (VL ≥ 1000 copies/mL), the majority (34.6%) were in the age group of 40–49, while among those virally suppressed (VL < 50 copies/mL), the majority (30.1%) were in the age group of 30–39 years (Figure 4a). Across the study area, viral load suppression rate was 75% (Figure 4b).

### 3.5. Identified Comorbid Conditions in the Study Population

Diabetes, hypertension, obesity, HBV infection, and dyslipidaemia are five of the HIV comorbid conditions identified in the study population (Table 2, Figure 5). About 390 (52%) participants were positive for HIV while 360 (48%) were negative. Of the other conditions, obesity showed a 42% prevalence, 29% were hypertensive, and 15% were diabetic (Table 2). Figure 5 presents identified comorbidities according to subdistricts; Elundini had the highest prevalence for hypertension (44.8%). The highest prevalence of diabetes was found in the Mhlontlo subdistrict at 42.3%, and the King Sabata Dalindyebo (KSD) subdistrict had the highest prevalence of HIV infection at 38.1% (Figure 5).

### 3.6. Prevalence of Dyslipidaemia

Of the four lipids analyzed, Triglycerides and HDL Cholesterol did not follow a normal distribution. Median and IQR were calculated in this case (Table 3 and Figure 6). Of the 750 participants, after elimination of duplicates and rejected specimens for serum lipids in the laboratory, 683 reported on total cholesterol, 680 on triglycerides, 676 on HDL cholesterol, and 647 on LDL cholesterol (Table 3). The overall prevalence of elevated triglyceride concentrations was 10.4%, with a cut-off of ≥2.3 mmol/L (Table 4). The overall prevalence of low HDL cholesterol concentrations was 82.7%, using a cut-off of <1 mmol/L. The overall prevalence of dyslipidaemia was 23% (95% CI 19.9–26.4) for elevated triglycerides, 82.7% (95% CI 79.6–85.5) for low HDL cholesterol, 16.1% (95% CI 13.4–19.1) for elevated total cholesterol, and 36% (95% CI 32.3–39.9) for elevated LDL cholesterol (Table 4). The prevalence of elevated LDL cholesterol was 45.8% in participants with hypertension and 42.5% in obese participants. The prevalence of elevated triglycerides was 41.7% in participants with diabetes and 31.1% in hypertensive ones. Low HDL cholesterol concentrations were observed in all disease conditions at 84.9% in HIV participants, 78.9% in diabetics, 84.7% in hypertensives, and 79.9% in obese participants (Figure 7).

## 4. Discussion

### 4.1. Principal Findings

In this study, we described the disease burden in rural districts of the Eastern Cape Province by assessing the prevalence of HIV and comorbid conditions in outpatients from randomly selected primary care settings. The study population was predominantly female at 86.5%; HIV, hypertension, diabetes, obesity, HBV infection, and dyslipidaemia were among the underlying medical conditions discovered. The rate of suppressed viral load was 75%; however, the age group of 40–49 years (34.6%) and the Elundini district (15.2%) had the highest nonsuppressed viral load. Our findings are consistent with data from the Institute of Health Metrics and Evaluation (IHME), which show that in South Africa, HIV/AIDS and cardiovascular diseases have remained among the top ten (10) most prevalent diseases over the last decade [25]. Interestingly, the burden of HIV/AIDS and cardiovascular diseases highlighted by our study findings reflects a similar trend in the profiles of top 10 diseases in Sub-Sahara Africa [26], and particularly countries with middle socioeconomic status, such as Cuba, Botswana, Equatorial Guinea, Gabon, Iraq, Namibia, and Sanin Lucia [25]. In addition, the most recent South African district health barometer report confirms that HIV and noncommunicable diseases (NCDs) are the most prevalent disease burdens nationally, regionally, and subregionally, particularly in the district examined in this study (OR Tambo and Joe Gqabi) [10,27,28].

### 4.2. Clinical Implications for Relevant Health Stakeholders and Policymakers

For decades, despite known nationwide coverage of antiretroviral therapy, South Africa has remained at the epicenter of the HIV/AIDS burden compared to other HIV-endemic settings [29,30]. Furthermore, the prevalence rate of over 50% for HIV infection in the study setting, which was rural, is concerning, especially since this rate is higher than the current, national HIV prevalence rate of approximately 13% [31]. This vast difference raises the question: what could be causing the high prevalence of HIV despite available interventions such as the ‘test and treat’ policy and antiretroviral therapy (ART)? Is it possible for this trend to be attributable to HIV under-reporting, or sociobehavioral factors in rural areas? Based on these findings, health stakeholders must improve the implementation of current intervention strategies and uptake in rural areas to reduce the high burden of HIV, exacerbated by the ongoing COVID-19 pandemic. Since the onset of COVID-19, institutional bodies such as the Council for Scientific and Industrial Research (CSIR) and the Albert Luthuli Centre have developed a COVID-19 Vulnerability Indicator Index which captures areas of high transmission rates and health susceptibility based on age and underlying disease conditions [27,28]. This COVID-19 Vulnerability Indicator Index is used even in the study districts that have reported many COVID-19 cases, but we are awaiting a documented report from this assessment. Furthermore, evidence from around the world, including South Africa, shows that people living with HIV have more severe outcomes and higher comorbidities from COVID-19 than those not living with HIV [32].

Aside from the HIV burden, scholars worldwide are noting an increase in the prevalence of NCDs in low- and middle-income countries, a phenomenon called the epidemiological transition, but the extent of this transition in the study region is unknown [26]. In addition, most countries in Sub-Saharan Africa lack established vital statistics systems and reliable population-level data, so data on the observed epidemiological transition from communicable diseases such as HIV and NCDs such as cardiovascular disease are limited. Nonetheless, the impact of this growing transition, which has resulted in an increase in NCDs in this region, has been discovered to pose significant challenges to healthcare systems [33]. In our study, the burden of NCDs included diabetes—14.7%, hypertension—28.9%, and obesity—42.4%. Additionally, as earlier mentioned, these NCDs were higher in the HIV-negative individuals compared to the HIV-positive individuals. Across the study population, low HDL cholesterol was most prevalent even among those with diabetes, hypertension, obesity, and HIV. In a recent meta-analysis on the burden of dyslipidaemia in Africa [9], high prevalence was found among adults in the continent and in South Africa. Therefore, we concur with institutional authorities, such as IHME GBD and the United Nations, that addressing risk factors such as chronic conditions, including HIV (attributable to ARTs), and dyslipidaemia, which could lead to noncommunicable diseases, will result in more robust health systems, healthier people, and greater resilience to COVID-19 and other future pandemic threats [34,35].

### 4.3. Strengths and Limitations

To our knowledge, the assessment of the prevalence of HIV and comorbid disease burden in patients attending selected primary health care facilities in rural districts of the Eastern Cape is the first in this region. One limitation beyond our control was that data on respiratory diseases such as TB, which is the most common comorbid infectious disease for people living with HIV, were not available at the time of data collection In addition, during the modification of the core and the expanded step-by-step WHO questionnaire for data collection, the focus was more on HIV and noncommunicable diseases. Furthermore, our analysis was mainly descriptive. Future studies should examine risk factors and demonstrate causality in the study setting, at the national level, and in other HIV endemic settings.

## 5. Conclusions

Our findings revealed that diseases such as HIV and cardiometabolic conditions were highly prevalent in the study setting, and we would like to encourage health stakeholders to strategize and implement interventions to reduce the burden of identified disease conditions, especially in rural areas. We place emphasis on rural areas because population health there tends to be overlooked due to disparities in the socioeconomic index, access to quality health care, and available resources compared to urban settings. Therefore, with the ongoing COVID-19 pandemic and in case of the emergence of any other disease in a manner similar to that of COVID-19, we recommend that health stakeholders in the study area and other geographical settings conduct periodic disease profile assessments to develop tailored interventions to improve healthy living and outcomes in line with the SDG and UHC agenda.

## Figures and Tables

**Figure 1 ijerph-19-08003-f001:**
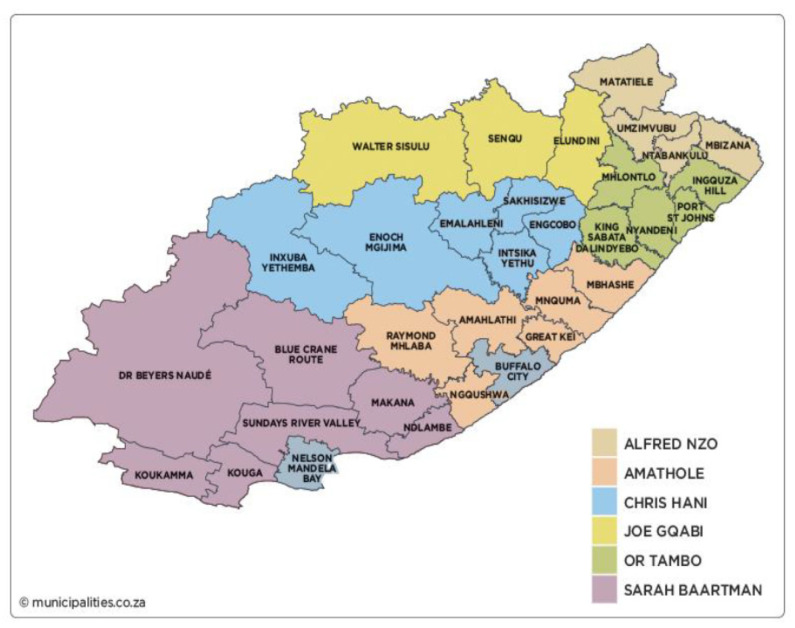
Map showing the study location.

**Figure 2 ijerph-19-08003-f002:**
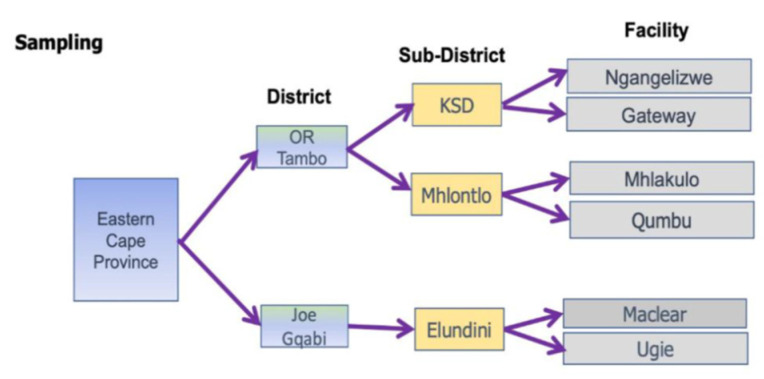
Study sampling and site.

**Figure 3 ijerph-19-08003-f003:**
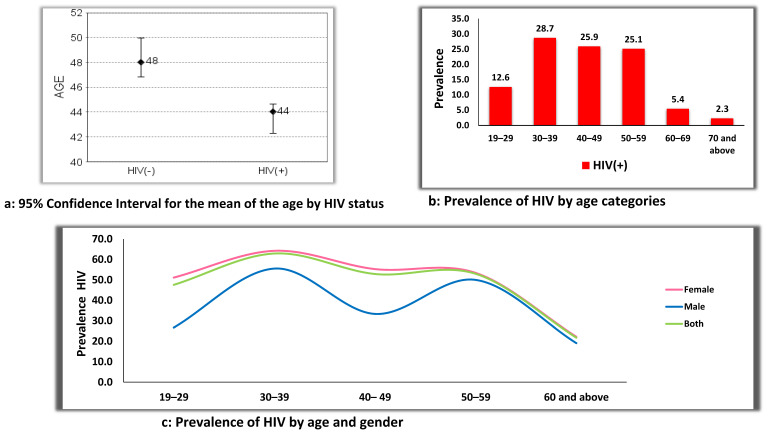
(**a**–**c**): Prevalence of HIV by age and gender.

**Figure 4 ijerph-19-08003-f004:**
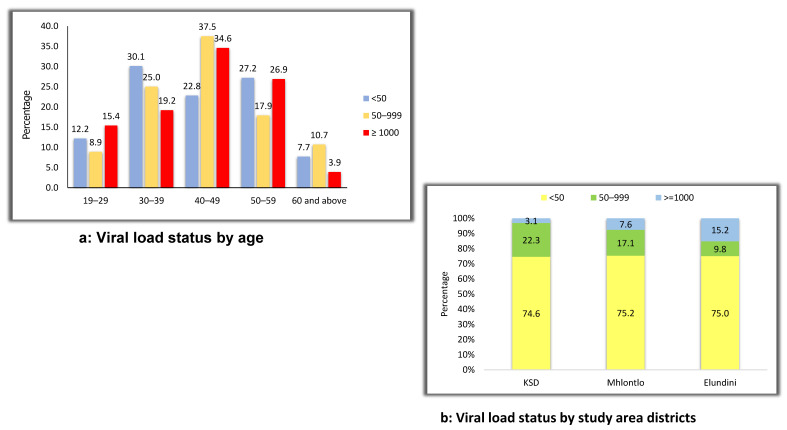
(**a**,**b**): Viral load status by age and study area.

**Figure 5 ijerph-19-08003-f005:**
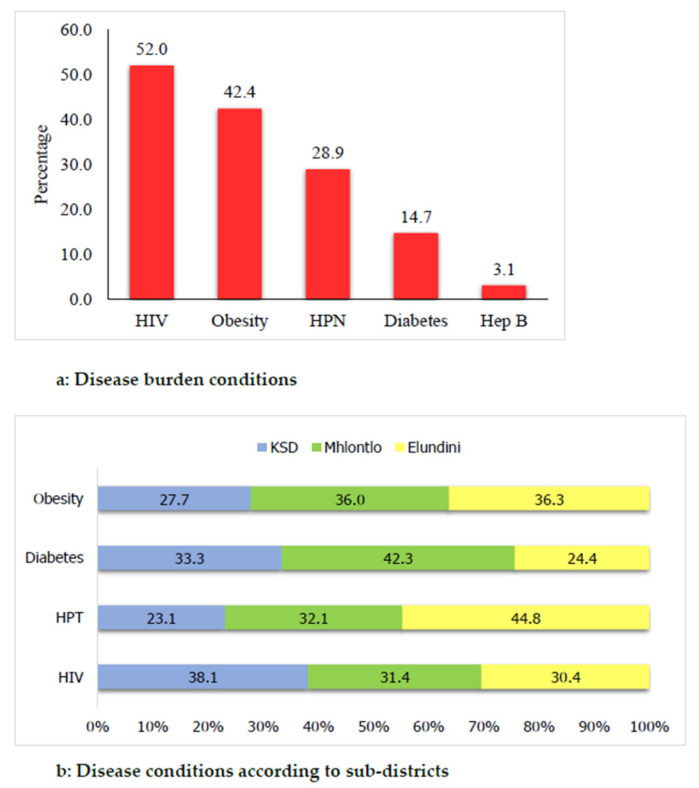
Prevalence of identified disease conditions.

**Figure 6 ijerph-19-08003-f006:**
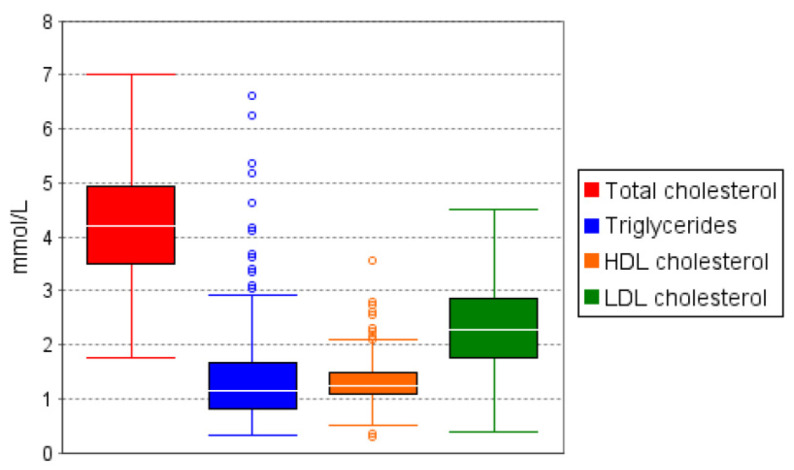
Boxplot for the lipids.

**Figure 7 ijerph-19-08003-f007:**
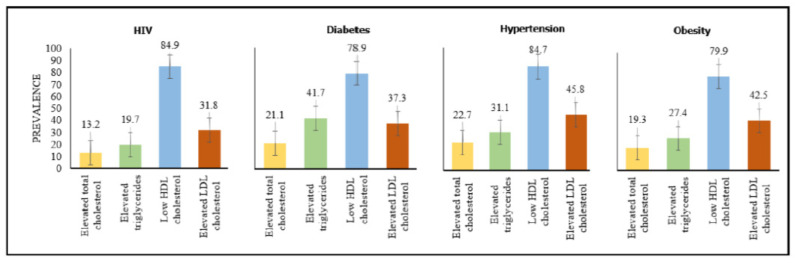
Pooled prevalence of dyslipidaemia in participants with HIV, diabetes, hypertension, and obesity. The cutoff for elevated total cholesterol was >6.18 mmol/L, elevated triglycerides were ≥2.3 mmol/L, reduced HDL cholesterol was <1 mmol/L, and elevated LDL was ≥4.1mmol/L.

**Table 1 ijerph-19-08003-t001:** Sociodemographic characteristics of the study population.

Sociodemographic Characteristics		Frequency	%
**Age group**	19–29	96	12.8
	30–39	173	23.1
	40–49	180	24.0
	50–59	175	23.3
	60 and above	126	16.8
**Gender**	Male	101	13.5
	Female	649	86.5
**Marital status**	Divorced	23	3.1
	Living together	20	2.7
	Married	263	35.1
	Single	374	49.9
	Widowed	70	9.3
**Educational attainment**	Illiterate	39	5.2
	Primary	180	24.2
	Secondary	434	58.3
	Tertiary	92	12.3
**Employment status**	Unemployed	428	57.1
	Employed	322	42.9
**Current smokers**	Smoker	56	7.6
	Not smoker	679	92.4

**Table 2 ijerph-19-08003-t002:** Identified disease conditions.

Comorbidity	Frequency	%
**HIV**	390	52.0
**Obesity**	316	42.4
**HPN**	217	28.9
**Diabetes**	73	14.7
**HBV**	22	3.1

HIV—Human immunodeficiency virus; HPN—Hypertension; HBV—Hepatitis B virus.

**Table 3 ijerph-19-08003-t003:** Lipid profile of the study population.

Lipids	N	Mean	Std. Dev	Median	IQR	95% CI
**Total Cholesterol**	683	4.22	0.97	-	-	[4.15, 4.30]
**Triglycerides**	680	-	-	1.15	0.86	[1.29, 1.41]
**HDL Cholesterol**	676	-	-	1.25	0.40	[1.28, 1.34]
**LDL Cholesterol**	647	2.32	0.97	-	-	[2.26, 2.38]

HDL—high-density lipoprotein; LDL—low-density lipoprotein.

**Table 4 ijerph-19-08003-t004:** Prevalence of lipids.

Lipids		Frequency	%
**Total Cholesterol (*n* = 683)**	<5.18 (Desirable)	573	83.9
	5.18–6.18 (Borderline high)	96	14.1
	>6.18 (High)	14	2.0
**Triglycerides (*n* = 680)**	<1.7 (Normal)	523	76.9
	1.8–2.2 (Borderline high)	86	12.6
	≥2.3 (High)	71	10.4
**HDL Cholesterol (*n* = 676)**	≥1 (Desirable)	117	17.3
	<1 (At risk)	559	82.7
**LDL Cholesterol (*n* = 647)**	<2.6 (Normal)	414	64.0
	2.6–4.0 (Borderline high)	222	34.3
	≥4.1 (High)	11	1.7

Note: Triglycerides and HDL Cholesterol did not follow a normal distribution. Median and IQR were calculated in this case.

## Data Availability

Data will be made available upon request from the corresponding author.

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
