# Peer review of "Prevalence of HIV and Selected Disease Burden in Outpatients of Primary Health Care (PHC) Facilities in Rural Districts of the Eastern Cape Province, South Africa"

_ijerph, 2022, doi:10.3390/ijerph19138003_

Round 1
Reviewer 1 Report
Not still ready for an international global audience, Scientific merit of work needs to be highlighted and the importance of the study made concise and impactful. English language edits required thoroughly.
Author Response
Response to Reviewer 1 Comments
Point: Not still ready for an international global audience, Scientific merit of work needs to be highlighted and the importance of the study made concise and impactful. English language edits required thoroughly.
Response (Please provide your response for Point above (in red):
We thank the reviewer for the overall general comments provided. We have reviewed our work and also engaged the services of an English language and scientific editors who has reviewed our manuscript as you recommended.

Reviewer 2 Report
In this study, the authors describe the disease burden in rural districts of the Eastern Cape Province, the poorest province in South Africa, by assessing the clinical profile of outpatients from randomly selected primary health care facilities (PHCF). Considering the lack of data on the prevailing disease burden in poor rural communities in South Africa this article provides valuable insight that should be used improve healthcare decisions. This study provides useful introspection and a powerful framework that should be expanded to other rural PHCF across Sub-Saharan Africa. These data would facilitate improved strategic allocation of resources.
Major
- Demographics of the study population (Table 1) should include the age stratification using the same intervals as figures 3 and 4. These data are necessary to properly evaluate the results shown in figure 4.
- Include the data to support this statement from the discussion (lines 215-217) “When the identified underlying conditions were compared by HIV status, only Diabetes and Hypertension were significantly higher among the HIV negatives than the HIV positives.”
Minor
- In the abstract change “non-viral load suppression" to “non-suppressed viral load”
- Once established use your acronyms, such as NCD, SDG, UHC, ECP etc.
- General copy editing required. eg typo on line 63 “conditions d by”
- Do the error bars in fig 3a indicate the 95% confidence interval? Include this information in the figure legend.
- Emphasize random selection of PHC facilities (line 211).
- The sentence on line 214 – 215 needs to be corrected. Patients in the 40-49 age range had a 34.6% prevalence while based on fig 4b the overall prevalence appears to be closer to 25%. Stating both results in the discussion would be useful.
- Be consistent, Hep B and HBV are both used.
Author Response
Response to Reviewer 2 Comments
Point 1:
Demographics of the study population (Table 1) should include the age stratification using the same intervals as figures 3 and 4. These data are necessary to properly evaluate the results shown in figure 4.
Response to Point 1 (Please provide your response for Point above (in red):
Thank you so much, we have revised Table 1 as directed.
Point 2:
Include the data to support this statement from the discussion (lines 215-217) “When the identified underlying conditions were compared by HIV status, only Diabetes and Hypertension were significantly higher among the HIV negatives than the HIV positives.”
Response to Point 2 (Please provide your response for Point above (in red):
Thank you for the comment, the statement is deleted as is no longer relevant
Point 3:
In the abstract change “non-viral load suppression" to “non-suppressed viral load”
Once established use your acronyms, such as NCD, SDG, UHC, ECP etc.
General copy editing required. eg typo on line 63 “conditions d by”
Do the error bars in fig 3a indicate the 95% confidence interval? Include this information in the figure legend.
Emphasize random selection of PHC facilities (line 211).
The sentence on line 214 – 215 needs to be corrected.
Response to Point 3 (Please provide your response for Point above (in red):
Thank you so much, we have revised as directed. We also engaged the services of English language and Scientific writing editor to revise and improve the quality of the manuscript.
Point 4:
Patients in the 40-49 age range had a 34.6% prevalence while based on fig 4b the overall prevalence appears to be closer to 25%. Stating both results in the discussion would be useful.
Response to Point 4 (Please provide your response for Point above (in red):
Thank you.
The results for the two figures address different things.
Fig 4a is VL by age and Fig 4b is VL by location
Fig 4a: In this table, the percentages were calculated by the total of each column. There were 26 patients with VL >1000 and the most frequent age group was 40-49 which represented 34.6% of the total.
Fig 4b: In this table, the percentages were calculated by the total of the row, e.g., Elundini had 92 HIV patients and 14 patients (15.2%) had VL >1000.
We have mentioned this as directed in the discussion.
Point 5:
Be consistent, Hep B and HBV are both used.
Response to Point 5 (Please provide your response for Point above (in red):
Thank you so much, we have revised accordingly and made sure HBV was used instead of Hep B.

Reviewer 3 Report
- Consider revising the title - a little misleading as it only considers few diseases - maybe burden of HIV ...
- Generalisation of the study - with 6
- Page 2, line 64-66 - consider revising the statement ... people to people
- Page 2 line 86, ref style
- Clinical profile is limited to HIV and co-morbidities
- the article is relevant however the title is misleading. Doing a clinical profile would mean they include all clinical services from the centers but the study results-focussed only on selected outcomes - HIV
Author Response
Response to Reviewer 3 Comments
Point 1, 5 & 6:
- Consider revising the title - a little misleading as it only considers few diseases - maybe burden of HIV
- Clinical profile is limited to HIV and co-morbidities
- The article is relevant however the title is misleading. Doing a clinical profile would mean they
include all clinical services from the centers but the study results-focused only on selected
outcomes - HIV
Response to Point 1, 5 & 6 (Please provide your response for Point above (in red):
Thank you so much, we have revised as directed by providing three possible options listed below and we are open to use the one most suitable and highly recommended by you.
Describing selected disease burden in outpatients of primary health care (PHC) facilities in rural districts of the Eastern Cape Province, South Africa
OR
Prevalence of HIV and selected disease burden in outpatients of primary health care (PHC) facilities in rural districts of the Eastern Cape Province, South Africa
OR
Profiling HIV and co-morbid disease burden in outpatients from selected primary health care (PHC) facilities in rural districts of the Eastern Cape Province, South Africa
Point 2, 3 & 4:
- Generalization of the study - with 6
- Page 2, line 64-66 - consider revising the statement ... people to people
- Page 2 line 86, ref style
Response to Point 2, 3 & 4 (Please provide your response for Point above (in red):
Thank you so much, we have revised as directed. We also engaged the services of English language and Scientific writing editor to revise and improve the quality of the manuscript.

Round 2
Reviewer 1 Report
Needs a lot of work and edits
Author Response
Your feedback is greatly appreciated; we have reviewed and revised our work accordingly.
Reviewer 2 Report
The authors addressed my initial concerns and I'm happy to recommend this manuscript be accepted for publication.
Author Response
Thank you so much for your review which has improved the quality and content of our manuscript.
Reviewer 3 Report
I would agree with your 2nd suggestion for title revision:
Prevalence of HIV and selected disease burden in outpatients of primary health care (PHC) facilities in rural districts of the Eastern Cape Province, South Africa
Author Response
Your review comments are greatly appreciated; we have reviewed and revised our work as directed. Thanks again